# RL in Name Only? Analyzing the Structural Assumptions in RL post-training for LLMs

## Abstract

Reinforcement learning based post-training of large language models (LLMs) has recently gained attention, particularly following the release of DeepSeek R1 (Guo et al., 2025), which applied GRPO for fine-tuning. Amid the growing hype around improved reasoning abilities attributed to RL post-training, we critically examine the formulation and assumptions underlying these methods. We start by highlighting popular structural assumptions made in modeling LLM training as an MDP, and show how they lead to a degenerate MDP that doesn't quite need the RL/GRPO apparatus. The two critical structural assumptions include (1) making the MDP states be just a concatenation of the actions with states becoming the context window and the actions becoming the tokens in LLMs and (2) splitting the reward of a state-action trajectory uniformly across the trajectory. Our comprehensive analysis demonstrates that, due to these simplifying assumptions, the standard approach is effectively equivalent to on-policy variant of outcome-driven iterative supervised learning. Our experiments on benchmarks including GSM8K and Countdown using Qwen-2.5 base models and Llama-3.2 instruct models show that Filtered Iterative SFT, incorporating both positive and negative samples, achieves performance comparable to GRPO-based training. We also show that these structural assumptions indirectly incentivize RL to generate longer sequences of intermediate tokens which in turn feeds into the narrative of "RL incentivizing thinking because it generates longer thinking traces." We continue to believe that RL writ large can be a great tool for post-training LLMs, and hope that our analysis of the limiting assumptions in the currently popular RL framework encourages work that goes beyond patching the symptoms of faulty assumptions and towards improving the underlying formulation.

## 1 Introduction

Recent advances in Large Language Models (LLMs) have substantially improved their ability to perform complex reasoning tasks, such as mathematical problem-solving. Typically, this is achieved through post-training techniques, including RL methods like Group Relative Policy Optimization (GRPO). These methods frame language modeling as a Markov Decision Process (MDP), where token generation is treated as a sequential decision-making task.

In this work, we critically examine the foundations of applying RL to LLM post-training through the lens of the specific MDP formulation that has been popularly used to model LLMs as agents (Guo et al., 2025). Under this formulation, each state is a sequence of previously generated tokens, transitions are deterministic concatenations of tokens, rewards are assigned only at the terminal state based on external verification of correctness, and the credit assignment problem is short-circuited by effectively dividing this terminal reward across all tokens (or actions) equally. These structural assumptions sharply distinguish the setting from the general MDP framework typically considered in RL, and drive several wrong intuitions.

Our contributions are twofold. Firstly, we theoretically show that under these structural assumptions, the full machinery of RL methods in LLMs post-training is somewhat redundant. By decoupling and simplifying the GRPO objective for trajectories labeled as positive (correct solutions) and negative (incorrect solutions), we demonstrate that on-policy variant of iterative supervised fine-tuning (SFT) guided by an external verifier yields comparable performance. This equivalence implies that RL

might not be imparting genuinely novel reasoning capabilities to the LLM, but rather leveraging latent capabilities inherent within the pretrained models.

Secondly, we empirically demonstrate that the terminal reward distribution assumption, wherein the relative advantage score is uniformly allocated across all tokens in a response, directly contributes to the increasing response lengths observed throughout training. Contrary to the claims made in the DeepSeek-R1 (Guo et al., 2025) paper, which attributes the increased response length at test-time to computational scaling, self-reflection, self-verification, and back-tracking, we show that the primary driver of response elongation is this uniform credit distribution. Since each token is incentivized equally without considering the "return-to-go" from any intermediate state, RL-based methods like GRPO inadvertently encourage verbose outputs.

This paper thus aims to clarify the misconceptions regarding the role of RL in enhancing LLM capabilities by providing a comprehensive theoretical and empirical analysis showing that idiosyncratic assumptions made in formulating the MDP influence model behavior during post training. A key implication of our analysis is that misunderstanding the consequences of these structural assumptions has led to research directions that attempt to fix superficial symptoms such as introducing length penalties (Arora & Zanette, 2025) or selectively filtering training data (Shrivastava et al., 2025) rather than addressing the root cause of length bias inherent in the basic formulation. We stress that our critique is directed at the specific MDP formulation currently used to model LLM post-training, rather than at RL as a general paradigm for LLMs post-training. This current, restrictive formulation does not fully realize the expressive power of RL. By highlighting these limitations, we aim to encourage the development of richer formulations that can further unlock the potential of RL in LLM post-training.

## 2 RELATED WORK

Supervised fine-tuning and reinforcement learning have emerged as the two main paradigms for post-training LLMs. Although early work focused heavily on SFT variants (Zelikman et al., 2022; Yuan et al., 2023; Dong et al., 2023; Gulcehre et al., 2023; Singh et al., 2023), recent efforts have shifted toward RL-inspired methods such as GRPO and its extensions (Liu et al., 2025; Zhang & Zuo, 2025; Yu et al., 2025; Yuan et al., 2025).

While the original SFT involved finetuning on the traces and solutions supplied externally, a line of work has explored improving language model reasoning via self-generated data, filtering data with verifier and iteratively fine-tuning on the filtered data, a process which we refer to as filtered iterative SFT in our work. Self-Taught Reasoner (STaR) (Zelikman et al., 2022) fine-tunes a model on its self generated data by iteratively producing rationales using greedy decoding. It retains only samples whose rationales lead to correct answers. Subsequent works have replaced greedy decoding with temperature sampling as an alternative data generation strategy. Rejection Sampling Fine-Tuning (RFT) (Yuan et al., 2023) collects reasoning paths by generating multiple outputs from a supervised model and filtering them using binary feedback (accepted/rejected). The model is then fine-tuned once on the accepted samples. RAFT (Reward-ranked Fine-Tuning) (Dong et al., 2023) introduces an iterative framework consisting of three steps: (1) generate a batch of outputs from the model; (2) score the outputs using a reward function and filter high-reward samples and (3) fine-tune the model on the filtered data.

Reinforced Self-Training (ReST) (Gulcehre et al., 2023) similarly generates multiple output sequences per prompt, scores them with a reward function, and fine-tunes the model using a reward-weighted loss, rather than hard filtering. ReST$^{EM}$ (Singh et al., 2023) extends this setup by fine-tuning the original base model in each iteration, instead of the latest model checkpoint. These methods collectively explore different ways to combine self-generated data, filtering, and reward signals for language model fine-tuning.

The recent and growing work has focused on variations of the GRPO Framework. GRPO (Shao et al., 2024) introduces group-relative advantages, avoiding state-based value estimation by directly assigning advantages based on relative scores within a group. Several follow-up works build on or modify GRPO in different ways. Dr-GRPO (Liu et al., 2025) removes standard deviation normalization and sample-length scaling from the GRPO loss, and points out that the objective may introduce a bias toward longer responses. GRPO-Lead (Zhang & Zuo, 2025) extends GRPO by adding a

length-dependent reward to encourage concise outputs, an explicit penalty to reduce low-precision responses, and an advantage reweighting strategy to handle limited rollout data.

In parallel, DAPO (Yu et al., 2025) introduces a different set of techniques for long chain-of-thought settings. It uses token-level policy gradient loss and includes four key components: Clip-Higher to maintain policy diversity, Dynamic Sampling to improve training stability, Overlong Reward Shaping to reduce noise from long outputs, and the removal of flat-reward samples from training. VAPO (Yuan et al., 2025) builds on DAPO by introducing value model pretraining, a decoupled and length-adaptive GAE, and group sampling to better handle sparse rewards and variable-length responses.

In this work, We argue that GRPO and its variants, despite their RL framing, rely on structural simplifications such as degenerate MDPs and uniform token-level reward assignment, which effectively reduce the actual learning process to iterative filtered supervised fine-tuning. A very recent work, RAFT++ (Xiong et al., 2025) also extended RAFT by incorporating importance sampling and clipping and shows comparable performance as GRPO.

Further, Thought Anchors Bogdan et al. (2025) conducted counterfactual analyses not all tokens contribute equally to producing the correct answer. Their work identifies pivotal tokens/sentences that are particularly influential in reaching the correct final answer and future sentences, using attention patterns. In contrast, GRPO assigns equal weight to all tokens, whereas traditional RL methods typically do a credit assignment based on cost to go. This makes GRPO effectively a form of RL in name rather than in execution.

There have also been conflicting claims about the response length: the longer responses have both been celebrated as a sign of improved reasoning Guo et al. (2025), and also viewed as a source of inefficiency, with several efforts directed at shortening them. Such efforts include sample-level loss (Yu et al., 2025; Liu et al., 2025), length-dependent penalties (Yuan et al., 2025; Aggarwal & Welleck, 2025; Arora & Zanette, 2025), and token-based control, where Thinkless (Fang et al., 2025) uses <think>/<short> tokens, while AdaptThink (Zhang et al., 2025), invokes <think> only on difficult questions. However, we show that increase in response length is more likely a side effect of the uniform advantage distribution followed with length scaling. Also, (Fatemi et al., 2025) shows theoretical analysis that generation of longer responses stems from RL-based optimization during training.

## 3 BACKGROUND

### 3.1 STANDARD MDP FORMULATION

A finite horizon Markov Decision Process (MDP) can be formally represented by a tuple $M = \langle S, A, P, r, \rho, H \rangle$. Here, $S$ is a finite set representing the state space, $A$ is a finite set representing the action space, and $P(s'|s, a)$ denotes the transition probability distribution from state $s$ to state $s'$ given action $a$. The scalar reward function $r(s, a)$ indicates the immediate reward received after performing action $a$ in state $s$, while $\rho(s_0)$ describes the initial state distribution. The term $H$ represents the finite time horizon over which the process evolves. The solution to such an MDP is given by a policy $\pi(a|s)$, which provides a distribution over actions for each state, guiding the decision-making process to maximize cumulative rewards.

### 3.2 LLM-MDP AND RL FOR LLMS

Let's now consider the popular MDP model used for LLMs introduced in DeepSeek-R1 paper (Guo et al., 2025), which extends the standard MDP Formulation by incorporating a verifier $V$ into the environment. The resulting formulation is defined as $M_{LLM} = \langle S, A, P, r, \rho, H, V \rangle$. In this scenario, the LLM acts as an agent interacting with an environment whose state space is the set of all possible token sequences up to a maximum length $H$, defined as $S = \bigcup_{k=0}^{H} \mathcal{V}^k$. with $\mathcal{V}$ representing a finite vocabulary of tokens. The action space $A$ is precisely this vocabulary $\mathcal{V}$, corresponding to the selection of individual tokens.

The state transition dynamics in this MDP is deterministic. Given a current state $s_h = (v_1, ..., v_{k-1})$ and an action $a_h = v_k$, the subsequent state $s_{h+1}$ is deterministically formed by appending $v_k$ to the existing sequence, thus $P(s_{h+1} = (v_1, ..., v_k)|s_h, a_h) = 1$.

Episodes in this MDP formulation correspond to the generation of complete token sequences. Each episode ends either upon the generation of a special end-of-sequence token indicating the completion of the response or upon reaching the predefined maximum sequence length $H$. The initial state $s_0$ represents the question prompt from a reasoning problem dataset.

Rewards in this formulation are defined explicitly through an external verifier. Upon episode termination, a reward of 1 is given only if the suffix of the terminal state $s_H$, representing the complete sequence generated by the LLM agent, contains the correct solution[1] to the given problem according to the external verifier; otherwise, the reward is zero.

**Structural assumptions of the LLM-MDP**: This LLM-MDP formulation adheres broadly to standard MDP properties, but introduces two structural assumptions.

1. **States as sequences of actions**: The state is represented as a sequence of tokens generated so far. This implies that each state inherently encodes the complete trajectory of actions (tokens). Hence, unlike standard MDPs, each state explicitly contains the historical context of actions taken to reach it from the initial state.

2. **Terminal reward and credit assignment**: Rewards are determined solely by an external verification at the terminal state, explicitly depending on the correctness of the final sequence. DeepSeek-R1 also shortcuts the credit assignment problem by splitting the reward (or any derived advantage metric, such as the group relative advantage used in GRPO) evenly across all tokens within the terminal sequence. This kind of reward assignment relaxes the conventional RL premise of making intermediate decisions based on expected future returns.

In this setting, fine-tuning an LLM is equivalent to improving the agent's policy $\pi_\theta$ to maximize the expected cumulative reward. RL policy gradient methods are well-suited for this task, as LLMs are transformer-based parameterized neural networks that output a probability distribution over the vocabulary given the context. In the next subsection, we provide an overview of GRPO (Shao et al., 2024), a policy gradient method used for fine-tuning LLMs.

### 3.3 GROUP RELATIVE POLICY OPTIMIZATION (GRPO)

Group Relative Policy Optimization (GRPO) is a RL algorithm introduced to reduce the computational cost of standard policy optimization methods, particularly in the context of fine-tuning large language models (LLMs). Traditional policy optimization methods, such as PPO, often require training a separate critic model to estimate the value function, which can be computationally expensive especially when the critic is as large as the policy model itself. GRPO avoids this overhead by estimating the advantage function using relative rankings within a group of sampled outputs, instead of relying on a separate value network.

Given a question $q$, GRPO samples a group of $G$ responses $\{o_1, \ldots, o_G\}$ from the old policy $\pi_{\theta_{\text{old}}}$. For each token $o_{i,t}$ in response $o_i$, the updated policy $\pi_\theta$ is trained to maximize the following objective:

$$\mathcal{J}(\theta) = \mathbb{E}_{q \sim P(Q), \{o_i\}_{i=1}^G \sim \pi_{\theta_{\text{old}}}(O|q)}$$

$$\frac{1}{G} \sum_{i=1}^{G} \frac{1}{|o_i|} \sum_{t=1}^{|o_i|} \left\{ \min\left( \frac{\pi_\theta(o_{i,t}|q, o_{i,<t})}{\pi_{\theta_{\text{old}}}(o_{i,t}|q, o_{i,<t})} \hat{A}_{i,t}, \text{clip}\left( \frac{\pi_\theta(o_{i,t}|q, o_{i,<t})}{\pi_{\theta_{\text{old}}}(o_{i,t}|q, o_{i,<t})}, 1-\varepsilon, 1+\varepsilon \right) \hat{A}_{i,t} \right) \right.$$

$$\left. - \beta \mathbb{D}_{KL}\left[ \pi_\theta \| \pi_{\text{ref}} \right] \right\}$$

$$(1)$$

Here, $\epsilon$ clipping factor is a hyperparameter, $\hat{A}_{i,t}$ is the standardized advantage for token $o_{i,t}$, computed from the group rewards $\mathbf{r} = \{r_1, r_2, \ldots, r_G\}$ as: $\hat{A}_{i,t} = \frac{r_i - \text{mean}(\mathbf{r})}{\text{std}(\mathbf{r})}$ and, a KL divergence penalty

---

[1] In practice, large language models (LLMs) often generate intermediate reasoning traces (e.g., between `<Think>` tags) followed by final answers (between `<Sol>` tags), where the `<Think>` section reflects chain-of-thought reasoning and the `<Sol>` section is evaluated by a verifier. However, as in the DeepSeek formulation, the state is defined as the sequence of tokens without distinguishing between reasoning and solution parts. We adopt the same simplification and treat the state as the just token sequence, irrespective of internal structure.

term is included to ensure that the updated policy does not deviate too far from a reference policy, with $\beta$ controlling the strength of this regularization. The KL divergence is computed at the token level as:

$$\mathbb{D}_{KL}\left[\pi_\theta \| \pi_{\text{ref}}\right] = \frac{\pi_\theta(o_{i,t})}{\pi_{\theta_{\text{old}}}(o_{i,t})} - \log\left(\frac{\pi_\theta(o_{i,t})}{\pi_{\theta_{\text{old}}}(o_{i,t})}\right) - 1$$

## 4 GRPO AS FILTERED ITERATIVE SUPERVISED FINE-TUNING

In this section, we show how the GRPO objective, under common structural assumptions made in the LLM-MDP formulation, simplifies to a form that closely resembles filtered iterative supervised fine-tuning (F-ISFT). We argue that when responses are assigned binary rewards via an external verifier and the resulting advantages are uniformly distributed across tokens, the GRPO update effectively reduces to weighted supervised learning over both positively and negatively labeled responses.

### 4.1 SIMPLIFYING THE GRPO OBJECTIVE

We begin with the GRPO objective:

$$\mathcal{J}(\theta) = \mathbb{E}_{q \sim P(Q), \{o_i\}_{i=1}^G \sim \pi_{\theta_{\text{old}}}(O|q)}$$

$$\frac{1}{G}\sum_{i=1}^G \frac{1}{|o_i|}\sum_{t=1}^{|o_i|}\left\{\min\left(\mathcal{ISR}_{i,t}(\theta)\hat{A}_{i,t}, \text{clip}\left(\mathcal{ISR}_{i,t}(\theta), 1-\varepsilon, 1+\varepsilon\right)\hat{A}_{i,t}\right) - \beta\mathbb{D}_{KL}\left[\pi_\theta\|\pi_{\text{ref}}\right]\right\} \tag{2}$$

where $\mathcal{ISR}_{i,t}(\theta)$ is the importance sampling ratio: $\mathcal{ISR}_{i,t}(\theta) = \frac{\pi_\theta(o_{i,t}|q,o_{i,<t})}{\pi_{\theta_{\text{old}}}(o_{i,t}|q,o_{i,<t})}$

Recent works ((Liu et al., 2025), (Yu et al., 2025), (Yuan et al., 2025)) show that the KL penalty term has limited effect on performance, since the clipping operation already keeps the updated policy within the trust region of the old policy ((Schulman et al., 2015), (Schulman et al., 2017)). Therefore, we can relax the KL penalty term from the objective, simplifying it to:

$$\mathcal{J}(\theta) = \mathbb{E}\left[\frac{1}{G}\sum_{i=1}^G \frac{1}{|o_i|}\sum_{t=1}^{|o_i|}\min\left(\mathcal{ISR}_{i,t}(\theta)\hat{A}_{i,t}, \text{clip}\left(\mathcal{ISR}_{i,t}(\theta), 1-\varepsilon, 1+\varepsilon\right)\hat{A}_{i,t}\right)\right] \tag{3}$$

Now let's assume that the importance sampling ratio $\mathcal{ISR}_{i,t}(\theta)$ are within the clipping range $(1-\varepsilon, 1+\varepsilon)$ (later we can put the clipping again), the objective further simplifies to:

$$\mathcal{J}(\theta) = \mathbb{E}\left[\frac{1}{G}\sum_{i=1}^G \frac{1}{|o_i|}\sum_{t=1}^{|o_i|}\mathcal{ISR}_{i,t}(\theta)\hat{A}_{i,t}\right] \tag{4}$$

Since the relative advantage $\hat{A}_{i,t}$ is computed at the output level and is constant across all tokens $o_{i,t}$ for a given response $o_i$, we can pull it out of the inner summation as $\hat{A}_{i,t} = \hat{A}_i$. This further simplifies the objective:

$$\mathcal{J}(\theta) = \mathbb{E}\left[\frac{1}{G}\sum_{i=1}^G \frac{\hat{A}_i}{|o_i|}\sum_{t=1}^{|o_i|}\mathcal{ISR}_{i,t}(\theta)\right] \tag{5}$$

### 4.2 DECOMPOSING BY POSITIVE AND NEGATIVE RESPONSES

Given a question $q$, we can divide its $G$ responses $\{o_1, o_2, ..., o_G\}$ sampled from the old policy $\pi_{\theta_{old}}$ into the group of positive responses $\mathcal{G}^+$ and group of negative responses $\mathcal{G}^-$ based on the binary reward value assigned to each responses from the external verifier. For every positive response of a given question $q$ the relative score will be the same and we denote it as $\hat{A}_q^+$ and the relative advantage score for negative responses will also be the same and we denote it as $\hat{A}_q^-$. Now we can split the objective function in positive trajectory and negative trajectory of given question $q$ and the objective equation is written as -

$$\mathcal{J}(\theta) = \mathbb{E}\left[\frac{1}{G}\left[\sum_{i=1}^{\mathcal{G}^+}\frac{\hat{A}_q^+}{|o_i|}\sum_{t=1}^{|o_i|}\mathcal{ISR}_{i,t}(\theta) + \sum_{i=1}^{\mathcal{G}^-}\frac{\hat{A}_q^-}{|o_i|}\sum_{t=1}^{|o_i|}\mathcal{ISR}_{i,t}(\theta)\right]\right] \tag{6}$$

We define $A_{q,i}^+ = \frac{\hat{A}_q^+}{|o_i|}$ and $A_{q,i}^- = \frac{\hat{A}_q^-}{|o_i|}$, allowing us to rewrite the objective function as:

$$\mathcal{J}(\theta) = \mathbb{E}\left[\frac{1}{G}\left[\sum_{i=1}^{\mathcal{G}^+} A_{q,i}^+ \sum_{t=1}^{H} \mathcal{ISR}_{i,t}(\theta) + \sum_{i=1}^{\mathcal{G}^-} A_{q,i}^- \sum_{t=1}^{H} \mathcal{ISR}_{i,t}(\theta)\right]\right] \quad (7)$$

The objective function derived above closely resembles that of filtered iterative supervised fine-tuning (F-ISFT). In standard F-ISFT, only positive responses are considered when updating the model parameters, with the goal of increasing the log-likelihood of their tokens. However, if we also incorporate negative responses into the objective, by increasing the log-likelihood for positive responses and decreasing it for negative ones, we obtain performance comparable to that of GRPO.

In this formulation, the terms $A_{q,i}^+$ and $A_{q,i}^-$ serve as adaptive weights for the positive and negative components of the objective, respectively. When rewards are binary, these weights which are computed from the mean and standard deviation of rewards within a group, become function of model's probability of success in findind a correct solution. Let p, be the probability of succeess. Specifically, Mroueh (2025) shows that higher weight is assigned to correct responses when $p < 0.5$ and to incorrect responses when $p > 0.5$. with equal weighting at $p = 0.5$. In contrast, FISFT assigns equal weights to correct and incorrect responses, in Equation 8.

The gradient of the above objective can be expressed using the identity $\nabla_\theta \pi(\theta) = \pi(\theta)\nabla_\theta \log \pi(\theta)$, which allows us to represents the gradient in terms of the log-probability of the tokens under the current policy.

$$\nabla_\theta \mathcal{J}(\theta) = \mathbb{E}\left[\frac{1}{G}\left[A^+ \sum_{i=1}^{\mathcal{G}^+} \sum_{t=1}^{H} \mathcal{ISR}_{i,t}(\theta)\nabla_\theta log(\pi_\theta(o_{i,t}|q, o_{i,<t}))\right.\right.$$
$$\left.\left. + A^- \sum_{i=1}^{\mathcal{G}^-} \sum_{t=1}^{H} \mathcal{ISR}_{i,t}(\theta)\nabla_\theta log(\pi_\theta(o_{i,t}|q, o_{i,<t}))\right]\right] \quad (8)$$

## 5 LENGTH BIAS IN GRPO

We observe that applying GRPO within the LLM-MDP framework induces a length bias in the learned model. Our empirical analysis further demonstrates that training on incorrect responses significantly contributes to this effect. Given a response with reward $r_i$ in a group $\mathcal{G}$, the group relative advantage is defined as: $A_i = \frac{r_i - \mu_\mathcal{G}}{\sigma_\mathcal{G}}$, where $\mu_\mathcal{G}$ and $\sigma_\mathcal{G}$ are the mean and standard deviation of rewards in the group.

**Correct responses** ($r_i > \mu_\mathcal{G}$, so $A_i > 0$): Distributing the positive advantage equally across tokens results in higher per-token rewards for shorter outputs, thereby incentivizing brevity.

**Incorrect responses** ($r_i < \mu_\mathcal{G}$, so $A_i < 0$): Distributing the negative advantage over more tokens lowers the per-token penalty, which encourages the generation of longer incorrect outputs.

This length bias arises from structural assumptions in the learning framework, specifically, dividing the advantage uniformly across tokens and scaling it with response length. We empirically show that this bias also appears in **F-ISFT+-**, similar to **GRPO**. Notably, **F-ISFT+-**, which is trained on both correct and incorrect responses, produces longer outputs than **F-ISFT+**, which is trained only on positive samples. Yet, the longer responses are often interpreted as signs of 'learning to reason' (Guo et al., 2025), even though they arise from disproportionately longer incorrect responses caused by structural assumptions in the degenerate LLM-MDP framework. We also provide further analysis on length bias in the Appendix A.

## 6 EXPERIMENTAL RESULTS

In this section, we present an experimental analysis supporting our hypothesis that, under the structural assumptions of the LLM-MDP formulation (see Section 3.2), Filtered-ISFT on both positive and negative outputs has the similar performance to that of GRPO. Furthermore, we empirically demonstrate that the observed increase in response length during GRPO training is a consequence of uniformly allocating the relative advantage score across all tokens in a response.

## 6.1 Experimental Setup

**Datasets**: We present our analysis on two datasets - 1) the *GSM8K* dataset (Cobbe et al., 2021), which is a widely used benchmark dataset consisting of grade school math problems, designed to evaluate the reasoning capabilities of large language models. It contains 8.5K problems, each paired with a question and an answer. The dataset is divided into 7.5K training problems and 1K test problems. And, 2) *Countdown* dataset (Countdown.), which is a generalized version of the classic 24 Game (Yang et al., 2022), where the objective is to combine a set of input numbers using basic arithmetic operations $(+, -, \times, \div)$ to reach a specified target number. In this dataset, each problem consists of 3 to 4 two-digit input numbers, with the target number also being a two-digit number. The dataset contains 9K examples, split into 8K training instances and 1K test instances. We also report the accuracy on the *MATH-500* dataset, a 500-problem test set evaluated using models trained on GSM8K.

**Models:** We conduct our experiments using two base models from the Qwen-2.5 family: (1) Qwen-2.5-0.5B and (2) Qwen-2.5-1.5B. and two base models from the Llama-3.2 family: (1)Llama-3.2-1B-Instruct and (2) Llama-3.2-3B-Instruct.

**Baselines:** We compare the performance of the following baselines in our experiments:

- **GRPO:** Group Relative Policy Optimization optimizes the objective function defined in Equation - 1 and updates the base language model parameters by reinforcing it with outputs sampled from the real-time policy model.
- **GRPO-wo-KL:** GRPO without the KL penalty term. In this baseline, we relax the KL divergence regularization from the GRPO objective, allowing us to evaluate GRPO's performance without this constraint. This is particularly relevant since the filtered iterative supervised fine-tuning baselines do not include a KL penalty.
- **Filtered-ISFT+:** Filtered Iterative Supervised Fine-Tuning using only positive outputs. In this baseline, the base model is trained on filtered positive responses sampled from the real-time policy by maximizing the log-likelihood of the tokens in these positive outputs.
- **Filtered-ISFT-:** Filtered Iterative Supervised Fine-Tuning using only negative outputs. In this baseline, the base model is trained on filtered negative outputs sampled from the real-time policy by minimizing the log-likelihood of the tokens in these negative outputs.
- **Filtered-ISFT+–:** Filtered Iterative Supervised Fine-Tuning using both positive and negative outputs. In this baseline, the base model is trained by increasing the log-likelihood of tokens in the filtered positive outputs and decreasing the log-likelihood of tokens in the filtered negative outputs, both sampled from the real-time policy model.

**Training Hyper-parameters**: For all baseline implementations, we use the VERL pipeline (Sheng et al., 2024). All baselines are trained with consistent hyperparameter settings to ensure fair comparison. The training batch size is set to 64 and the mini-batch size to 8 for both datasets. To ensure an equal number of training samples across baselines, we sample 5 responses per question prompt for each method. During response rollouts, the temperature is set to 0.6. The maximum prompt length is set to 512 for GSM8K and 256 for countdown, while the maximum response length is fixed at 1024 for both datasets. The learning rate is set to $1e-6$. For GRPO, we set the KL divergence coefficient to $\beta = 1e-3$. In the case of Filtered-ISFT+–, we apply a constant weighting of 0.5 to both positive and negative responses throughout training. Model checkpoints are saved every 10 steps for evaluation on the test set. All experiments on the GSM8K dataset are conducted using a single A100 GPU, while Countdown experiments are run using two H100 GPUs. For the GSM8K dataset, both models of the Qwen and Llama family are trained for 145 global time steps, corresponding to 5 epochs. For the Countdown dataset, Qwen-2.5-0.5B is trained for 600 global time steps and Qwen-2.5-1.5B & Llama-3.2-3B-Instruct are trained for 300 global time steps. For the larger language models (DeepSeek-Math-7B-Instruct and Qwen3-8B) trained on GSM8K, we keep all hyperparameters the same as detailed above, except that instead of full-parameter fine-tuning, we apply LoRA fine-tuning for both models with rank 16 and alpha 32. The training is performed on two H100 GPUs for both the larger model for a total of 200 global training steps.

**Implementation Complexity and Training Dynamics**

The implementation complexity of Filtered-ISFT+-, in terms of both space and computation, is roughly the same as GRPO. Similar to GRPO, at each training iteration we sample a batch of question

prompts from the dataset and generate n responses per question prompt from the current model policy and label each response as positive or negative based on the correctness using an external verifier, store them in a buffer, and perform multiple gradient updates on that buffer. The updates simply increase the log-likelihood of tokens in positive responses and decrease the log-likelihood of tokens in negative responses.

Further in implementation, similar to GRPO, F-ISFT variants do not consider groups in which all responses are correct or all responses are incorrect. These cases correspond to empirical estimates of the base policy's probability of success being 1 and 0, respectively. Mroueh (2025) provides a detailed analysis showing that these cases p = 0 & 1, correspond to fixed-point iterations of the objective function and therefore do not change the policy. Consequently, we filtered out groups containing only correct or only incorrect responses. In addition, Xiong et al. (2025) performed ablation studies on these groups and observed a decrease in performance when included.

In terms of training dynamics, both methods achieve similar performance, but F-ISFT+- demonstrates greater stability compared to GRPO. As shown in Equation 8, F-ISFT+- assigns equal weight to correct and incorrect responses during training, whereas GRPO uses adaptive weights based on empirical estimates of the mean and variance under the given policy, which depends on sampling

FISFT effectively functions as an on-policy variant of SFT. With the additional inclusion of importance sampling for on-policy correction, clipping, and the filtering of groups where all responses are correct or incorrect, it becomes FST in name only as much as GRPO is RL in name only. We make the comparison that, under structural assumptions, GRPO becomes equivalent to a modified form of SFT that iteratively trains on self-generated data, incorporating modifications such as clipping, importance sampling, and group filtering when all responses are either correct or incorrect

## 6.2 RESULTS AND DISCUSSION

**Performance of GRPO *vs.* F-ISFT variants** – Table 1 shows the performance comparisons of all baselines for both Qwen-2.5 models (0.5B and 1.5B) on the GSM8K and Countdown datasets. On GSM8K, all baselines achieve comparable performance, and both Qwen-2.5-0.5B and Qwen-2.5-1.5B show substantial gains after post-training. Filtered-ISFT+-, which incorporates both positive and negative examples, exhibits training dynamics and test performance closely aligned with GRPO. Similarly, Filtered-ISFT+, trained only on positive samples, achieves performance comparable to both GRPO and Filtered-ISFT+-. On the Countdown dataset, greater variance is observed across baselines. GRPO without KL regularization shows lower performance, possibly due to increased exploration during training. While both base models start with low accuracy, post-training significantly improves their performance. On the smaller model, Filtered-ISFT+ slightly outperforms GRPO, whereas on the larger model, GRPO achieves slightly better results.

| Methods | Qwen-2.5-0.5B | | Qwen-2.5-1.5B | |
|---|---|---|---|---|
| | GSM8K | CD8K | GSM8K | CD8K |
| Base Model | 00.07 | 00.05 | 22.67 | 00.09 |
| GRPO | **55.87** | 37.73 | **78.24** | **71.43** |
| GRPO-wo-KL | 55.19 | 42.43 | 76.87 | 70.82 |
| Filtered-ISFT+ | 51.71 | **44.01** | 74.98 | 53.57 |
| Filtered-ISFT- | 55.27 | 00.00 | 76.04 | 00.00 |
| Filtered-ISFT+- | 55.72 | 37.86 | 76.37 | 65.89 |

Table 1: Performance of all baselines on test dataset of GSM8k and CD8k for both Qwen-2.5 family models (0.5B and 1.5B). For each dataset, the highest score is highlighted in **bold**.

| Methods | Deepseek-math-7B-Instruct | Qwen3-8B |
|---|---|---|
| | GSM8K | GSM8K |
| Base Model | 00.07 | 65.27 |
| GRPO | 82.4 | **92.1** |
| PPO | 81.9 | 91.8 |
| Filtered-ISFT+- | **83.7** | 91.5 |

Table 2: Performance of GRPO, PPO and Filtered-ISFT+- on test dataset of GSM8k for Deepseek-math-7B-Instruct and Qwen3-8B models. The highest score is highlighted in **bold**.

As a sanity check, we also evaluated Filtered-ISFT–, which uses only negative samples during fine-tuning. While its performance on GSM8K was on par with other baselines, it significantly underperformed on Countdown. This suggests that negative-only supervision may be inadequate for more challenging tasks such as Countdown, where positive examples likely play a critical role in guiding learning and improving generalization.

Table 2 shows the performance of GRPO, PPO, and F-ISFT+- on the GSM8K dataset for the larger models *DeepSeek-Math-7B-Instruct* and *Qwen3-8B*. From the results, it is evident that our F-ISFT± method performs comparably to GRPO even at these larger scales. This further supports our claim that the behavior we analyze is not limited to small models and persists across stronger base models as well. Additionally, both GRPO and F-ISFT+- achieve similar performance to PPO while being more scalable, as they do not require a value function.

| Methods | *Llama-3.2-1B-Instruct* | | | *Llama-3.2-3B-Instruct* | | |
|---|---|---|---|---|---|---|
| | GSM8K | MATH-500* | CD8K | GSM8K | MATH-500* | CD8K |
| Base Model | 35.78 | 23.40 | 0.00 | 75.13 | 40.80 | 00.01 |
| GRPO | 62.01 | **33.02** | 0.00 | **84.59** | 34.00 | 55.96 |
| Filtered-ISFT+ | 57.31 | 25.40 | 0.00 | 81.95 | 48.60 | **56.92** |
| Filtered-ISFT+- | **63.07** | 30.20 | 0.00 | 83.54 | **49.40** | 54.91 |

Table 3: Performance of all baselines on the test sets of GSM8k, MATH-500, and CD8k for the LLaMA-3.2 Instruct models (1B and 3B). For each dataset, the highest score is highlighted in **bold**. Note that results on the MATH-500 dataset are obtained using models trained on the GSM8k dataset.

We further evaluated the performance of all baselines on additional language models - the Llama-3.2 Instruct family models (1B and 3B) on GSM8K, Countdown, and MATH500, with results reported in Table 3. Experiments with Llama-3.2-1B-Instruct and Llama-3.2-3B-Instruct show that F-ISFT+- and GRPO perform similarly on GSM8K and CD8K. On MATH500, GRPO performs better on the 1B model, while F-ISFT+- achieves slightly higher performance on the 3B model. Although F-ISFT+ has comparatively lower performance on the 1B model, all baselines achieve similar performance across datasets on the 3B model.

These results demonstrate that GRPO and Filtered-ISFT+- perform similarly. Thus, our empirical analysis show that under the given structural assumptions, RL in the current LLM-MDP framework is effectively equivalent to F-ISFT incorporating both positive and negative samples.

**Length bias analysis** - The figures in 1 present results on the Countdown test dataset at different evaluation steps during the post-training of the Qwen-2.5-1.5B model. The left panel shows the number of correct and incorrect responses, where we observe that the number of correct responses increases steadily while the number of incorrect responses decreases across all baselines. The right panel shows the average token response length for both correct and incorrect responses. For correct responses, we find a slight increase in the average response length for GRPO and Filtered-ISFT+-, while it remains relatively constant for Filtered-ISFT+. For incorrect responses, the average length decreases initially but then increases substantially for GRPO and Filtered-ISFT+-, whereas Filtered-ISFT+ maintains comparable response lengths for both correct and incorrect responses.

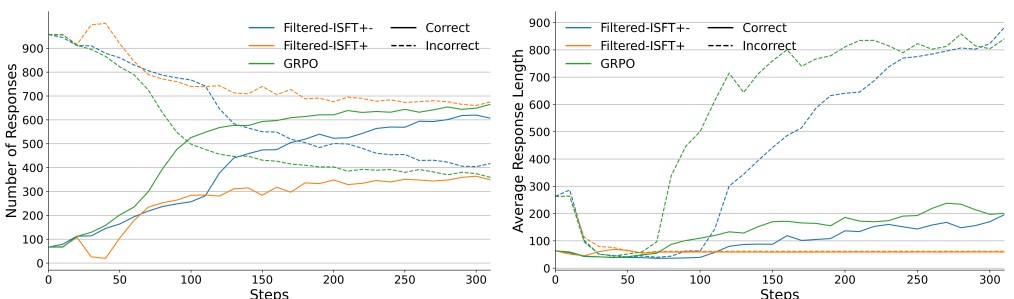

Figure 1: Left: Number of correct and incorrect responses. Right: Average response lengths for correct and incorrect responses, on the Countdown test dataset at each evaluation step during post-training of the Qwen-2.5-1.5B base model using GRPO and variant of Filtered-ISFT algorithm.

These length-based analyses indicate that Filtered-ISFT+- also incentivizes longer outputs, often producing even longer responses than Filtered-ISFT+. This suggests that the inclusion of incorrect responses during training contributes to increased output length, similar to the behavior observed

in GRPO. We attribute this to RL's inherent bias toward longer responses as discussed in Section 5 which arises from the uniform distribution of advantages across tokens combined with length scaling. Empirically, we observe that training on incorrect responses leads to increased response length, as the loss continues to decrease with increasing response length on these examples.

# 7 DISCUSSION AND CONCLUSION

In this work, we critically examined the structural assumptions underlying the popular LLM-MDP formulation and analyzed how they undermine RL applications to LLM post-training. Focusing on the widely adopted GRPO algorithm introduced by DeepSeek (Guo et al., 2025), we showed that by representing states as nothing more than sequences of actions, applying only terminal rewards, and uniformly distributing credit, GRPO closely resembles filtered iterative supervised fine-tuning. Our empirical results further demonstrated that variants of filtered iterative SFT, leveraging both positive and negative samples, achieve performance comparable to GRPO. We also observed that RL-trained LLMs tend to generate longer responses, an effect often misattributed to improved reasoning capabilities. Our findings suggest that this phenomenon actually arises from biases in the training setup, specifically the uniform distribution of advantages across tokens combined with length-based scaling effects.

A key implication of our analysis is that misunderstanding the consequences of these structural assumptions has led to a number of downstream research directions that attempt to correct for symptoms of the formulation, rather than addressing the root cause. For example, recent works have proposed explicitly optimizing for shorter reasoning traces by introducing length penalties in the reward function (Arora & Zanette, 2025; Team et al., 2025) or by avoiding training on longer incorrect responses indirectly, e.g. by sorting the responses by length and correctness (Shrivastava et al., 2025). While these methods appear to mitigate the tendency of RL-trained LLMs to produce excessively long outputs, they do so without recognizing that such length biases are intrinsic to the degenerate reward allocation mechanism of the current LLM-MDP framework. As a result, these interventions risk obscuring the fundamental issue and may unintentionally hinder the development of more principled approaches.

A promising direction may be to revisit the MDP formulation itself. The 2-LLM framework described in (Valmeekam et al., 2025) provides one such avenue. In this formulation, a chain-of-thought (CoT) LLM policy generates intermediate reasoning steps as actions conditioned on the problem prompt, while a main LLM generates the final solution conditioned on both the prompt and the CoT actions. This setup more faithfully reflects the sequential decision-making structure assumed in RL and enables the design of reward functions that meaningfully shape intermediate reasoning. By adopting such alternatives, the full expressive capacity of RL can be more effectively leveraged in guiding LLMs toward better reasoning strategies.

To conclude, our critique is directed at the degenerate LLM-MDP formulation rather than at RL itself. Under the current restrictive assumptions, the potential of RL cannot be fully realized. We hope that our analysis motivates exploration of alternative formulations that can better unlock the benefits of RL for LLM post-training.

## REPRODUCIBILITY STATEMENT

All source code is provided as supplementary material, accompanied by instructions to ensure ease of reproducibility.

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

# A   LENGTH BIAS

To strengthen our argument on response length, we distinguish between the think part and the solution part of a response. Since the verifier evaluates only the solution, the think part is never checked, and the overall accuracy is not influenced by its validity, even though the solution is conditioned on it. Thus, when we refer a response as long or short responses, we can treat the solution parts as equivalent and attribute length differences to the think part.

We find that, for a particular question, all correct responses share the same positive advantage, and all incorrect responses share the same negative advantage. The distinction arises from the length of these responses, as GRPO scales the final loss with response length. We reformulate this as scaling the advantage by the length of the response. As a result, longer responses, even if correct, may receive a smaller per-token learning signal, and vice versa.

RL aims to maximize accuracy through the correct solution part by selectively increasing the probability of tokens in the think part that contribute to correct solutions. These tokens could be part of either long or short responses but received higher advantages. Given this, the think part can be viewed as an prompt augmentation. Let us consider the length bias in the think part of responses with respect to correct and incorrect solution parts.

## A.1   LENGTH BIAS IN CORRECT SOLUTIONS WITH DIFFERENT THINK PARTS

Consider a think part of length $L_x$ with a correct solution, and another sequence of length $L_y > L_x$ that has also generated the correct solution part. Both receive the same reward, but their advantage per token differs because of length scaling. The shorter think part receives a higher advantage per token than the longer think part, due to length scaling. Similarly the solution tokens receive a higher advantage for the shorter think part than for the longer think part.

Now, let us analyze the conditional probabilities of solution parts given the different think parts.

The total loss can be decomposed into two parts: the **think** tokens and the **solution** tokens. (We analyze the simplified loss without clipping, as clipping limits updates and does not affect our analysis.)

Let $L_x = T_x + S_x$ be the total length of sequence $x$, where:

- $T_x$ is the number of "think" tokens (e.g., prompt augmentation)
- $S_x$ is the number of "solution" tokens

Then the loss for Shorter sequence $x$ is:

$$\mathcal{L}_x = -\frac{A_i}{L_x} \left( \sum_{t=1}^{T_x} \log \pi(o_t^{\text{think},x}|o_{<t}^x, q) + \sum_{t=T_x+1}^{L_x} \log \pi(o_t^{\text{sol},x}|o_{<t}^x, q) \right)$$

Similarly, for a longer sequence $y$ with $L_y = T_y + S_y$:

$$\mathcal{L}_y = -\frac{A_i}{L_y} \left( \sum_{t=1}^{T_y} \log \pi(o_t^{\text{think},y}|o_{<t}^y, q) + \sum_{t=T_y+1}^{L_y} \log \pi(o_t^{\text{sol},y}|o_{<t}^y, q) \right)$$

We know that $L_x < L_y$

$$\frac{A_i}{L_x} > \frac{A_i}{L_y}$$

Since the solution parts are nearly identical, $o_t^{\text{sol},x} \approx o_t^{\text{sol},y} \approx o_t^{\text{sol}}$,

Consider the conditional probabilities of sol $\pi(o_t^{\text{sol}}|o^{\text{think},x}, q)$, $\pi(o_t^{\text{sol}}|o^{\text{think},y}, q)$ with shorter think parts and longer think parts respectively.

The posterior probability of solutions with shorter think part receives a stronger training signal compared to longer responses. Consequently, training increases the conditional probability assigned to correct solutions with fewer reasoning tokens. Thus, the model gradually favors shorter reasoning steps within correct responses.

However, as the length of the think part approaches zero, meaning "no think part", the RL objective tends to maximize the conditional probability of solutions without reasoning due to higher advantages. Yet, the initial likelihood of producing correct solutions without think part may be low, particularly for harder or unsolvable problems compared to solvable ones. Consequently, as illustrated in Figure 2, we observe that the average response length decreases when the model encounters solvable problems but increases with unsolvable problems.

Thus, the think part emerge based on the initial likelihood and prior probabilities assigned by the base model, influencing the final solution depending on problem difficulty. We demonstrate that RL consistently favors shorter correct responses over longer ones. However, we highlight that depending on the difficulty of the problem, it is unlikely the shortest feasible response would entirely lack a reasoning component. Instead, some minimal prompt augmentations or reasoning steps typically persist.

## A.2 LENGTH BIAS IN INCORRECT RESPONSES WITH ATLEAST ONE CORRECT SOLUTION

Similarly, consider a prompt augmentation or think part of length $L_x$ that results in an incorrect solution, and another think part of length $L_y$ with $L_y > L_x$ that also leads to an incorrect solution. Due to length-based scaling, both responses receive different scaled advantage values per token. We assume that there exists another response of length greater than $L_y$ that yields a correct solution; otherwise, the advantage values would vanish.

We know that $L_x < L_y$, and

$$\frac{A_i}{L_x} < \frac{A_i}{L_y}, \quad \text{as } A_i \text{ is negative}$$

With similar argument as above, consider the conditional probabilities/posterior of solution tokens, $\pi(o_t^{\text{sol}}|o^{\text{think},x}, q)$, $\pi(o_t^{\text{sol}}|o^{\text{think},y}, q)$ with shorter and longer think parts respectively.

The posterior probability of an incorrect solution with a shorter response receives a stronger negative signal compared to a longer response with incorrect solution. This leads to a decrease in the

conditional probability of generating solutions with shorter think parts, thereby encouraging the model to produce longer think parts. For incorrect responses, scaling the advantage by length further contributes to an overall increase in response length. We present empirical evidence supporting this behavior below. Notably, this signal to generate longer reasoning segments is reinforced only when at least one correct response exists among the set of generated responses.

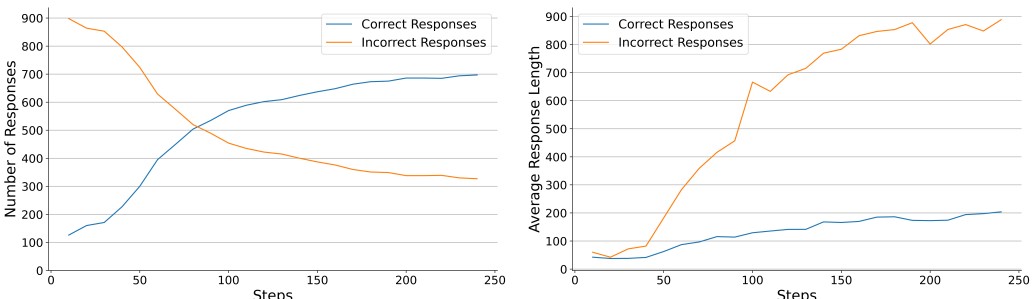

Figure 2: Left: Average response lengths for correct and incorrect responses. Right: Number of correct and incorrect responses, on the Countdown test dataset at each evaluation step during post-training of the Qwen-2.5-1.5B base model using the GRPO algorithm.

### A.3 LENGTH BIAS - EXPERIMENTAL RESULTS

Figure 2 presents an empirical analysis of response length and number of correct vs incorrect samples during RL post-training of the Qwen-2.5-1.5B model using GRPO. The left side of the figure shows the average response lengths for correct and incorrect samples, while the right side shows the number of correct and incorrect responses throughout training.

These observations support our analysis from Sections A.1 and A.2. Specifically, the average length of incorrect responses increases as training progresses, whereas the length of correct responses remains relatively stable. Moreover, the number of correct responses consistently rises, while the number of incorrect responses declined. Thus, we analyze the increase in average response length by separating it into contributions from correct and incorrect responses. Our findings show that the increase is mainly caused by the incorrect responses. This can be explained by the structural assumptions, particularly by uniform advantage distribution across tokens and scaling the advantage by length during training. Moreover, we find no evidence that longer responses lead to better reasoning and that the reasoning abilities could emerge due to increased response length

