# OpenReview forum: "RL in Name Only? Analyzing the Structural Assumptions in RL post-training for LLMs"
_ICLR.cc/2026/Conference — Submitted to ICLR 2026_

### Official Review · Reviewer_WZko · 2025-10-27

**Soundness:** 4
**Presentation:** 3
**Contribution:** 1
**Rating:** 2
**Confidence:** 5

**Summary:**

This paper critically analyzes the structural assumptions of reinforcement learning (RL)-based post-training for large language models (LLMs), especially focusing on Group Relative Policy Optimization (GRPO) popularized by DeepSeek R1. It identifies two key flawed assumptions in the dominant LLM-Markov Decision Process (MDP) formulation: states defined as concatenations of past tokens and uniform allocation of terminal rewards across all tokens. Theoretically, the paper shows the standard approach
is effectively equivalent to outcome-driven supervised learning; empirically, using datasets like GSM8K and Countdown with Qwen-2.5 and Llama-3.2 models, this SFT variant achieves performance comparable to GRPO. It also reveals that longer outputs from RL-trained LLMs are biases from the structural assumptions rather than improved reasoning, and calls for better MDP formulations to fully unlock RL’s potential for LLM post-training

**Strengths:**

1. The paper addresses a fundamental and timely question about the validity of current RL-based post-training formulations for large language models. By dissecting the structural assumptions that underlie widely used frameworks such as GRPO, the work provides valuable conceptual clarity that is highly relevant to the ongoing debate around “RL for reasoning.”

2. The analysis is thorough and well-grounded, connecting theoretical insights with empirical evidence on standard benchmarks (e.g., GSM8K, Countdown) and multiple base models.

**Weaknesses:**

1. It is unclear what is meant by ``RL in Name Only''. The term needs clearer definition or justification.
2. Questionable claim that RL-based training is equivalent to outcome-driven supervised learning.


While the paper argues that the MDP formulation degenerates under common assumptions, this does not fully justify equating RL with supervised learning. The critical distinction lies in on-policy versus off-policy optimization — which directly affects importance sampling in gradient estimation. A contemporaneous work by XiongW et al. reaches a more nuanced conclusion: outcome-driven supervised learning performs slightly worse than GRPO but remains competitive due to its simplicity. This aligns with the authors’ own training curves in arxiv version, which also show a modest but consistent performance gap favoring GRPO.

3. The discussion about sequence length lacks causal evidence.

The paper suggests that RL encourages longer reasoning traces because of the advantage. In GRPO-like algorithms, the loss is distributed across tokens proportionally to their advantages (e.g., averaged over sequence length) as shown in eq. 8, rather than being solely related to \frac{A_i}{L_x}. Therefore, changes in output length cannot be causally linked to the claimed mechanism without further controlled analysis.

[1]. Xiong W, Yao J, Xu Y, et al. A minimalist approach to llm reasoning: from rejection sampling to reinforce[J]. arXiv preprint arXiv:2504.11343, 2025.

**Questions:**

Please see question in section Weaknesses.

---

> ### Author Response · Authors · 2025-11-24
>
> > **It is unclear what is meant by ``RL in Name Only''. The term needs clearer definition or justification.**
>
> Regarding the phrase “**RL in Name Only,**” we use this to characterize the implementation of reinforcement learning in the widely publicized **DeepSeek R1**, particularly GRPO framework. Under the structural assumptions outlined in the paper, this approach yields a degenerate MDP. Specifically, the use of uniform credit assignment, wherein each token is assigned the same advantage and the state is defined as the concatenation of the action and previous state, renders the method **RL in name but not in execution**, as it fails to capture the cost-to-go associated with each state.
>
> > **Questionable claim that RL-based training is equivalent to outcome-driven supervised learning. While [..]**
>
> We are afraid the reviewer misunderstood the FISFT formulation. FISFT is also effectively on-policy (like GRPO), as shown in *Section 4.2.* It generates new data from the policy at the previous step and discards the data after each iteration. This is unlike off-policy methods, which are trained on the same static datasets over multiple iterations. The importance sampling ratio correction in FISFT, similar to GRPO, corrects for the rollouts sampled with respect to the policy from the previous step, mainly to stabilize training and to ensure that the updates remain on-policy. Thus our Filtered Iterative SFT variants, as the name suggests, iteratively samples new data and are on-policy, essentially similar to GRPO.
>
> We reiterate that GRPO, being the RL method with verifiable rewards, checks for final answer correctness and is driven primarily by binary outcomes under structural assumptions similar to outcome driven supervised learning. Both theoretically and empirically, we validate our claim that RL for LLM MDPs, particularly GRPO with binary outcome rewards, is equivalent to outcome-driven supervised learning.
>
> > **A contemporaneous work by XiongW et al. reaches a more nuanced conclusion: outcome-driven supervised learning performs slightly worse than GRPO but remains competitive due to its simplicity.**
>
> The proposed methods in Xiong et al., particularly **RAFT++** and **Reinforce-Rej.** differ from the FISFT variants that we derived and have underperformed compared to GRPO. **RAFT++** considers only positive samples and does not apply filtering when all samples in a group are correct. In contrast, our **FISFT+**, similar to **Reinforce-Rej**, filters out cases where all samples are correct, and is shown to slightly underperform **GRPO** similar to Reinforce-Rej. Thus our **FISFT+** is different from **RAFT++** variant, which hasn’t performed well compared to GRPO.
>
> Our **FISFT+-**, can be seen as extending  Reinforce-Rej, by including training on negative samples. In particular, it assigns equal weights to correct and incorrect samples while filtering out samples when all are either correct or incorrect. Thus FISFT+-, trained on both positive and negative samples shows performance comparable to GRPO. These findings are consistent with our additional experimental results on larger models now included in Table 2, along with implementation details *(lines 375-399)*.
> From results reported in **Table 1, 2 and 3,** it is clear that our experimental studies are theoretically well motivated. Our systematic evaluation of FISFT variants demonstrates that FISFT+- achieves performance comparable to GRPO across model families, sizes, and datasets. While the SFT and Reinforce variants in Xiong et al., mostly motivated from the ablation studies on different components of advantage estimation, underperformed relative to GRPO.
>
> > **The discussion about sequence length lacks causal evidence. The paper suggests that RL encourages longer reasoning traces.[..]**
>
> We respectfully request the reviewer to look at **Structural Assumption 2, uniform credit assignment**. In GRPO-like algorithms, the loss is distributed across tokens proportionally to their advantages, but since the **advantages are equal for each token,** A_i is equal for every token. So, we show theoretically in *Section 5* that for correct responses, the addition of new tokens would increase the loss.  In contrast, for incorrect responses, the addition of a new token would decrease the loss, and hence for incorrect responses, the length increases significantly. Therefore, the increase in response length is attributed to training on negative samples. We provide experimental evidence in **Figure 1**, showing that **FISFT+**, trained only on *positive samples*, does not show an increase, while **FISFT+-**, trained on both *positive and negative samples*, shows a length increase mainly attributed to training on negative samples. This clearly provides causal evidence that sequence length increases with training on incorrect responses, causally linked to the structural assumption of crediting every token with equal advantage in addition to length scaling.

---

> > ### Comment · Reviewer_WZko · 2025-11-26
> >
> > ## Your Explanation About Q2 “RL-based training is equivalent to outcome-driven supervised learning,”
> >
> > I want to clarify the distinction between on-policy and off-policy algorithms. On-policy reinforcement learning estimates the value of a policy using data generated by that same policy, meaning the actions used for exploration and learning are sampled from the current policy being improved. In contrast, off-policy learning estimates the value of a target policy while generating data from a different behavior policy.
> > Therefore, even when importance sampling is applied, **the method remains off-policy as long as the data were collected from a behavior policy different from the target policy.**
> >
> > ## Some Question about explanation on Q3
> > Do you think the method you proposed should outperform GRPO? Since it does not introduce any new data, I’m wondering how it is able to learn new knowledge. And based on your arXiv version **v2**, the performance also appears to be almost the same.

---

> > > ### Author Response · Authors · 2025-11-26
> > >
> > > >**Do you think the method you proposed should outperform GRPO? Since it does not introduce any new data, I’m wondering how it is able to learn new knowledge. And based on your arXiv version v2, the performance also appears to be almost the same.**
> > >
> > > Firstly, we would like to respectfully correct a misimpression the reviewer seems to have about our paper: Outperforming GRPO was never the main focus or intention of the paper. Our main focus was to analyse GRPO, the widely popularised RL method by Deepseek R1 for post training LLMs.
> > > We show that because of the structural assumptions, especially the one involving effectively dividing the outcome reward equally into all the tokens–including the intermediate tokens, DeepSeek R1 algorithm is in effect close to an on-policy variant of SFT that we called FISFT. We support this empirically also by showing that their performance is quite similar.
> > > Our paper shows that understanding this alternative lineage of the R1’s RL formulation puts into clearer perspective other idiosyncrasies of and confusions about the algorithm– such as attempts to fix length increase with length penalties when the effect is primarily caused by training on incorrect responses.
> > >
> > > >**Your Explanation About Q2 “RL-based training is equivalent to outcome-driven supervised learning,”
> > > I want to clarify the distinction between on-policy and off-policy algorithms. On-policy reinforcement learning estimates the value of a policy using data generated by that same policy, meaning the actions used for exploration and learning are sampled from the current policy being improved. In contrast, off-policy learning estimates the value of a target policy while generating data from a different behavior policy. Therefore, even when importance sampling is applied, the method remains off-policy as long as the data were collected from a behavior policy different from the target policy.**
> > >
> > > We previously addressed the distinction between on-policy and off-policy methods in the context of importance sampling for gradient estimation with respect to this particular comment:
> > > >**The critical distinction lies in on-policy versus off-policy optimization — which directly affects importance sampling in gradient estimation**
> > >
> > > While there is also a distinction based on the behavior policy and the target policy used in calculating those importance sampling ratios, we would like to point out that it does not apply here, as we are dealing with proximal policy methods. These methods such as TRPO, PPO, and GRPO are commonly referred to as on-policy methods because the policy used to sample responses is kept close to the current policy, particularly via the KL constraint in TRPO and the clipping mechanism in PPO. Therefore, GRPO is considered on-policy even though the samples are collected from the previous policy.
> > >
> > > We define FISFT as on-policy because it also applies clipping to importance sampling ratios (see section 4.2, eq 8). It does not train on samples collected from an arbitrary behavior policy stored in a buffer. Instead, similar to GRPO, it relies only on data generated by the policy from the previous iteration, which is kept proximally close to the current policy via clipping the importance sampling ratios. Following the same convention used for GRPO where the old and new policies remain close due to clipping and data are discarded after each iteration, FISFT is therefore treated as an on-policy method.

---

### Official Review · Reviewer_wmmp · 2025-10-30

**Soundness:** 3
**Presentation:** 3
**Contribution:** 3
**Rating:** 6
**Confidence:** 3

**Summary:**

This work examines the structural assumptions underlying reinforcement learning methods for LLM post-training, specifically focusing on GRPO as used in DeepSeek-R1. The authors argue that two key assumptions in the LLM-MDP formulation (representing states as token sequences and uniformly distributing terminal rewards) reduce GRPO to filtered iterative supervised fine-tuning. The contributions are as follows. First, the authors present a theoretical analysis showing how GRPO simplifies to weighted supervised learning under these assumptions. Second, they provide empirical evidence through experiments on GSM8K and Countdown datasets using Qwen-2.5 and Llama-3.2 models, demonstrating that F-ISFT with positive and negative samples achieves comparable performance to GRPO. Third, they show that increased response length during RL training stems from structural assumptions rather than improved reasoning.

The strengths of the paper are as follows. First, the paper addresses a timely concern about what RL accomplishes. Second, the mathematical derivation effectively demonstrates how structural assumptions lead to equivalence with F-ISFT. The step-by-step simplification is easy to follow. Third, there is a comprehensive experimental setup, across several model families and sizes. Finally, the paper tackles the root cause, rather than proposing another patch like length penalties, the paper identifies fundamental formulation issues, which is  valuable long-term.

The weaknesses of the paper are as follows. First, the largest model size explored was 3B, and findings may not hold for larger model sizes. Second, there is no comparison between RL methods using proper credit assignment e.g. MTCS.

**Strengths:**

The strengths of the paper are as follows.
- First, the paper addresses a timely concern about what RL accomplishes.
- Second, the mathematical derivation effectively demonstrates how structural assumptions lead to equivalence with F-ISFT. The step-by-step simplification is easy to follow.
- Third, there is a comprehensive experimental setup, across several model families and sizes.
- Finally, the paper tackles the root cause, rather than proposing another patch like length penalties, the paper identifies fundamental formulation issues, which is intellectually honest and valuable long-term.

**Weaknesses:**

The weaknesses of the paper are as follows.
- First, the largest model size explored was 3B, and findings may not hold for larger model sizes.
- Second, there is no comparison between RL methods using proper credit assignment e.g. MTCS.

**Questions:**

n/a

---

> ### Author Response · Authors · 2025-11-24
>
> > **W1 - Additional Results**
>
> We have now  added additional results on larger models DeepSeek-LLM-7B and Qwen3-8B, post-trained  on the GSM8K dataset (see Table 2 and line 427-431). As shown in the below reported table, our F-ISFT± method performs comparably to GRPO even at these larger scales. This further supports our claim that the behavior we analyze is not limited to small models and persists across stronger base models as well.
> | Dataset-GSM8k      | Deepseek-MATH-7B-Instruct | Qwen3-8B |
> |-------------|---------------------------|-----------|
> | Base Model  | 0.07                      | 65.27     |
> | GRPO        | 82.4                      | 92.1      |
> | PPO         | 80.3                      | 91.8      |
> | F-ISFT+-    | 83.7                      | 91.5      |
>
> > **W2 - Second, there is no comparison between RL methods using proper credit assignment e.g. MTCS.**
>
> In our paper, we primarily focus on analyzing the structural assumptions in the LLM–MDP formulation used in the DeepSeek R1 paper [1]. Even in their work, the authors acknowledge that RL methods incorporating proper credit assignment, such as MCTS-based approaches can, in principle, provide more accurate credit allocation. However, these methods are not scalable and are computationally expensive for LLM post‑training (see DeepSeek R1 [1], Guo et al., Section 4.2). This is because token generation presents an exponentially large search space, and MCTS relies on a pretrained value function to guide the search. Training such a fine‑grained value function is inherently difficult and is process reward models are prone to reward hacking.
>
> Therefore, in our paper, we argue that the structural assumptions in the LLM–MDP formulation cause GRPO to effectively reduce to an outcome‑driven supervised learning update that closely resembles F‑ISFT+-. These same assumptions also incentivize the model to generate longer sequences to reduce the per‑token penalty for incorrect responses (see Appendix A for a complete proof and detailed empirical analysis of GRPO’s length‑bias mechanism).
>
> Additionally, in the table above, we report PPO results on the GSM8K dataset. PPO learns a separate value function and performs credit assignment using that value estimate during training. However, the results clearly show that GRPO and F‑ISFT+- achieve similar performance to PPO while also being more scalable, since they do not require learning a value function.
>
> References:
> 1. Guo, et al. Deepseek-r1: Incentivizing reasoning capability in llms via reinforcement learning. arXiv:2501.12948, 2025.

---

### Official Review · Reviewer_SgUC · 2025-10-31

**Soundness:** 1
**Presentation:** 3
**Contribution:** 1
**Rating:** 2
**Confidence:** 5

**Summary:**

This paper investigates the two critical structural assumptions in current RL for LLMs. Authors's analysis demonstrates that, due to these simplifying assumptions, the standard approach is effectively equivalent to outcome-driven supervised learning. Authors also show that these structural assumptions indirectly incentivize RL to generate longer sequences of intermediate tokens. To support their claims, authors conducted experiments on GSM8K and Countdown using Qwen-2.5 base models and Llama-3.2 instruct models.

**Strengths:**

1. This paper is clearly written and easy to follow.

**Weaknesses:**

1. First of all, the authors claim that "Our comprehensive analysis demonstrates that, due to these simplifying assumptions, the standard approach is effectively equivalent to outcome-driven supervised learning". However, the validity of this claim requires further consideration. Although the derived Equation (8) looks like SFT, the expectation is taken over the distribution of the current policy, whereas SFT is done over a static dataset. RL learns over its own rollouts!
2. Limited technical novelty. The authors attribute the length bias of GRPO to its "average loss over the entire sequences", which has already been investigated in Section 3.3 of DAPO, the ByteDance's work about 5 months ago. Moreover, the authors claim that "we show that the primary driver of response elongation is this uniform credit distribution". However, in Figure 7(a) of DAPO's paper, we can still see that the response length increases during training, even though DAPO fixes the length bias in GRPO by Token-Level Policy Gradient Loss.

**Questions:**

Please refer to Weaknesses above.

---

> ### Author Response · Authors · 2025-11-24
>
> > W1 - First of all, the authors claim that "Our comprehensive analysis demonstrates that, due to these simplifying assumptions, the standard approach is effectively equivalent to outcome-driven supervised learning". However, the validity of this claim requires further consideration. Although the derived Equation (8) looks like SFT, the expectation is taken over the distribution of the current policy, whereas SFT is done over a static dataset. RL learns over its own rollouts!
>
> We believe the reviewer misunderstood the FISFT formulation. We refer the reviewer to Section 4.2, particularly in Equation (8) of FISFT, which shows that the expectation is indeed taken over the old policy, i.e., the policy from the previous step. As the name “Filtered Iterative SFT” suggests, the data is collected iteratively from the old policy, similar to GRPO, which is also updated from data collected using the policy from the previous step. Thus we define Iterative Filtered SFT as fine tuning performed iteratively on data generated by the model from the previous step. Also, the SFT dataset need not be static, as we pointed out in the related work (Section 2). Prior works such as STaR, RFT, RAFT, REST, and REST-EM have also performed SFT on self-generated data.
> Our theoretical analysis shows that, under the structural assumptions imposed by the LLM-MDP formulation, GRPO reduces to an outcome-driven supervised learning update that closely matches F-ISFT. Empirically, we further demonstrate that GRPO exhibits performance similar to F-ISFT trained on both correct and incorrect responses.
>
> > W2 -  Limited technical novelty. The authors attribute the length bias of GRPO to its "average loss over the entire sequences", which has already been investigated in Section 3.3 of DAPO, the ByteDance's work about 5 months ago. Moreover, the authors claim that "we show that the primary driver of response elongation is this uniform credit distribution". However, in Figure 7(a) of DAPO's paper, we can still see that the response length increases during training, even though DAPO fixes the length bias in GRPO by Token-Level Policy Gradient Loss.
>
> **Technical novelty compared to DAPO**. DAPO (Yu et al.) is contemporaneous work. In their paper, the authors argue that GRPO employs a sample-level loss formulation in which each sample’s loss is normalized or inversely scaled by its token length. As a result, longer sequences contribute less to the overall loss than shorter ones. Based on this observation, DAPO claims that GRPO fails to sufficiently penalize undesirable patterns that appear in long responses, and they only show empirically that this leads to increases in entropy and response length during training.
>
> In contrast, our paper provides both **theoretical analysis and empirical evidence** showing that GRPO actively incentivizes longer incorrect responses and shorter correct responses. (See Appendix A for a complete proof and detailed empirical analysis of GRPO’s length-bias mechanism.) Importantly, **DAPO does not analyze length-bias behavior separately for correct versus incorrect responses, whereas our work directly addresses this distinction.**
>
> **Length bias in DAPO**. Contrary to the claim that DAPO “fixes” GRPO’s length bias, the token-level policy-gradient loss introduced in Section 3.3 (Equation 12) does not eliminate the issue. From the equation, it is clear that length normalization is applied uniformly within each group of responses to a given prompt
> This means that, for a particular prompt, all tokens in correct responses irrespective of length of response (whether they belong to a longer or shorter response) are promoted equally with all tokens in incorrect responses that are suppressed irrespective of length. However DAPO continues to exhibit increasing response lengths during training, which we attribute to Structural Assumption 2, the uniform credit assignment. Since each token receives the same advantage and in this case the same length normalization factor, regardless of whether it belongs to longer or shorter responses, the addition of an extra token decreases the loss for incorrect responses. This explains that training with negative responses leads to an increase in output length for DAPO, as also observed in their Figure 7(a). Our experimental results show that FISFT+, trained only on positive responses, does not exhibit a significant increase in length, even without an explicit length penalty as in DAPO. In contrast, FISFT+-, trained on both correct and incorrect responses, shows an increase in length similar to GRPO.

---

### Official Review · Reviewer_UE3A · 2025-11-03

**Soundness:** 4
**Presentation:** 3
**Contribution:** 3
**Rating:** 6
**Confidence:** 3

**Summary:**

This paper breaks down structural assumptions in RL post-training for LLMs and GRPO and shows that under these assumptions, the GRPO objective can be simplified to a filtered iterative SFT objective. To prove this, the authors relax the KL term in the GRPO objective and then decompose it by positive and negative responses which results in a F-ISFT+- objective with both positive and negative responses incorporated. Empirical evidence shows that the F-ISFT+- trained models are comparable in performance to GRPO trained models on reasoning benchmarks.

It also provides an alternate explanation for the length bias in GRPO which suggests that the uniformly distributed terminal reward incentivizes the model to produce longer sequences to reduce the per-token penalty for incorrect responses. This is further supported by F-ISFT+- producing longer responses like GRPO but F-ISFT+ does not.

**Strengths:**

- The analysis and breakdown of the MDP assumptions and terminal reward assignment in RL LLM post-training was meticulous and led to very interesting conclusions
- The explanation for increasing response lengths in GRPO is compelling and opens up a new avenue for exploration to fix the issue with more sophisticated solutions than naive length penalties.
 - Quantitative results on Qwen and Llama models and GSM8K, Countdown and MATH datasets validate the theoretical claims in the paper and providing strong evidence for the central arguments.

**Weaknesses:**

- Empirical results are only shown on smaller (0.5B to 3B) models. Adding additional results with larger models would add further support to claims of the paper.
- A discussion on the implementation complexity and training dynamics of F-ISFT+- would improve the paper.
 - The removal of the KL penalty could be an over-simplification. Similarly, the assumption of binary rewards also might not hold true for all tasks, and F-ISFT is not necessarily comparable to other RL algorithms like PPO.
- While revisiting the MDP formultion is proposed as a direction to leverage the full capacity of RL, no setup is explored. This leaves the paper as critique with no proposed solutions.

**Questions:**

Please see weaknesses.

---

> ### Author Response · Authors · 2025-11-24
>
> >**W1 - Additional Results**
>
> We have now  added additional results on larger models **DeepSeek-LLM-7B** and **Qwen3-8B**, post-trained  on the **GSM8K dataset** *(see Table 2 and line 427-431)*. As shown in the below reported table, our F-ISFT± method performs comparably to GRPO even at these larger scales. This further supports our claim that the behavior we analyze is not limited to small models and persists across stronger base models as well.
> | Dataset-GSM8k      | Deepseek-MATH-7B-Instruct | Qwen3-8B |
> |-------------|---------------------------|-----------|
> | Base Model  | 0.07                      | 65.27     |
> | GRPO        | 82.4                      | 92.1      |
> | PPO         | 80.3                      | 91.8      |
> | F-ISFT+-    | 83.7                      | 91.5      |
>
> >**W2 - Implementation complexity and training dynamics of F-ISFT+-**
>
> The implementation complexity of F-ISFT+-, in terms of both space and computation, is roughly the same as GRPO. Similar to GRPO, at each training iteration we sample a batch of question prompts from the dataset and generate n responses per question prompt from the current model policy and label each response as positive or negative based on the correctness using an external verifier, store them in a replay buffer, and perform multiple gradient updates on that buffer. The updates simply increase the log-likelihood of tokens in positive responses and decrease the log-likelihood of tokens in negative responses. In terms of training dynamics, both methods achieve similar performance, but  F-ISFT+- demonstrates stability compared to GRPO, As shown in Eq (8), F-ISFT+- assigns equal weight to correct and incorrect responses during training, whereas GRPO uses adaptive weights based on empirical estimates of the mean & variance under the given policy, which depends on sampling. We added this discussion on implementation complexity and training dynamics in the paper (lines 375-399).
>
> > **W3 - The removal of the KL penalty ... other RL algorithms like PPO.**
>
> We would like to note that it  has become common practice to remove the KL penalty, and its removal has been shown not to affect performance. In fact, [1] shows that RL with verifiable rewards–as is the case with GRPO that employs rule-based verifiers–eliminates concerns about distributional shift between rewards estimated from a reference model and those from the current model. This is because unlike  RLHF, where regularization prevents the policy from deviating too far from the distribution on which the reward model was learnt (i.e, the reference policy), GRPO relies on verifiers rather than learned reward models. Thus, [2] also removed the KL term to reduce memory and computational overhead for reference policy. As reported in Table 1, GRPO achieves similar performance with and without the KL term on GSM8K.
>
> **Assumption of binary rewards**: Given the huge popularity of RL in Deepseek R1, GRPO and numerous incremental extensions built upon it, we are interested in analysing this particular setting, which only considers tasks with binary rewards.
>
> **Comparable to other RL algorithms like PPO**: Additionally, in the table above, we report the results of PPO on the GSM8K dataset. The results clearly show that GRPO and F-ISFT± have similar performance as PPO while also being more scalable, as they do not require a value function.
>
> > **W4 - While revisiting the MDP .. as a critique with no proposed solutions.**
>
> Our paper’s primary focus is on dissecting the structural assumptions in the LLM-MDP formulation used in the DeepSeek R1 paper and on demonstrating the limitations and side effects that arise from this degenerate MDP formulation - for example, showing that the GRPO objective becomes equivalent to F-ISFT+- and exposing the resulting length-bias issue. We further highlight that, without recognizing the limitations of this degenerate MDP formulation, much of the subsequent work has introduced piecemeal extensions to the GRPO algorithm aimed at mitigating symptoms of the formulation rather than addressing the root cause. Our analysis provides significant explanatory power by placing these otherwise isolated extensions to R1’s RL pipeline into a coherent conceptual context.
>
> In the future work section, we also outline one promising direction that we are currently pursuing: a 2-LLM framework. In this formulation, a chain-of-thought (CoT) LLM policy generates intermediate reasoning steps as actions conditioned on the problem prompt, while a main LLM produces the final answer conditioned on both the prompt and the CoT actions. Essentially the CoT LLM trained to generate prompt augmentations that help the main LLM to output the correct solution.
>
> References:
> 1. Hu, Jingcheng, et al. *Open-reasoner-zero: An open source approach to scaling up reinforcement learning on the base model.* arXiv:2503.24290 (2025).
> 2. Liu, Zichen, et al. *Understanding r1-zero-like training: A critical perspective.* arXiv:2503.20783 (2025).

---

### Author Response · Authors · 2025-11-24
**Summary of Changes in Revised Submission**

We thank all the reviewers for their thoughtful comments and valuable feedback. While we respond to each reviewer individually, we summarize the main changes in the revised version below. All the additions and modifications are done in blue.

**Additional Experiments on Large Models** : We included experiments on **Deepseek-Math-7B-Instruct** and **Qwen3-8B** using the **GSM8K Dataset** in **Table 2**. The results show that our claims scale across model sizes.

**Added New Baseline: PPO**: We include **PPO** as a new baseline in our additional experiments along with **FISFT ±** and **GRPO** (Table 2). Although PPO uses a value function, it has not shown improvement over GRPO.

**Implementation and Training Details**: We provide a detailed discussion on implementation of **FISFT** and **GRPO** (lines 375–399). These details clarify that FISFT functions as an on-policy variant of SFT.

**Updated Related Work**: We included the latest work, **Thought Anchors** (lines 122–127), which shows that not all tokens contribute equally and that certain pivot sentences are influential in determining the model’s final answer.

**In summary**, we added experimental results on large models and included a new baseline. We included the implementation and training details in the main paper and updated the relevant work with the latest works. We made the flow clear to emphasize that **GRPO is RL in name only** and that **FISFT functions as an on-policy variant of SFT,** to make readers appreciate the degeneracy in the LLM MDP formulation and the structural assumptions.

---

### Meta-Review · Area_Chair_HbWZ · 2025-12-29

**Summary:**

This paper received diverse review comments, half rejection and half acceptance, reflecting a divide on the validity and novelty of the authors' claims. I will detail the concerns from the reviewers later, but I have some personal opinions about this work, as follows, which lead to the decision that the paper is recommended to be rejected (that is to say, the rejection is not purely based on the negative scores from some reviewers).

This paper does a great job mathematically deriving the equivalence between GRPO and Filtered SFT (specifically F-ISFT), but I feel its conceptual framing limits its impact.

1. **It Creates a "False Negative" Narrative**: I think the paper falls into a semantic trap by framing its discovery as a "debunking" of RL (implying "It's not real RL, it's just SFT"). From my perspective, this is the wrong lens. Since we established that Filtered SFT is essentially 0-order RL (a search-based optimization), the paper shouldn't claim GRPO is "RL in name only." Better Framing: Instead of dismissing GRPO as "fake RL," the paper could celebrate the finding that "In the context of Language Models, RL naturally and optimally collapses into On-Policy Filtered SFT." This turns a negative critique into a powerful unifying theory.

2. **It Relies on a Narrow Definition of RL**: I feel the paper’s argument hinges on a rigid definition of RL--likely assuming that "Real RL" requires complex value functions, non-trivial state transitions, or temporal difference learning. However, I view Contextual Bandits and 0-order optimization as legitimate subsets of RL. The Missed Opportunity: The paper argues that because the MDP is "degenerate" (concatenated history), it's not RL. I think it would be more insightful to argue that LLM post-training is a Contextual Bandit problem, and therefore, methods like GRPO should look like Filtered SFT. The "degeneracy" is a feature, not a bug.

3. **It Oversells the "SFT" Label**: By reducing GRPO to "Outcome-Driven SFT," I think the paper glosses over the critical "Online" and "Negative Feedback" components we discussed.

- Standard SFT is static and positive-only.

- The "F-ISFT" they describe (On-policy sampling + Filtering) involves exploration (the E-step in EM). Conclusion: I believe calling this process "SFT" is theoretically reductionist. It obscures the fact that the model is actively searching the solution space, which is the soul of Reinforcement Learning, regardless of the label we attach to it.

Summary: I think the paper is mathematically sound but philosophically conservative. It spends too much energy arguing about labels ("Is it RL or SFT?") rather than focusing on the mechanism.

**Reviewer Concerns:**

### Concerns Addressed by the Rebuttal

- **Experimental scope (Model size)**
- **Missing baselines (PPO)**
- **Implementationdetails**
- **Related work**

### Remained or Disputed Concerns

- **Validity of "RL = SFT" Theory:** Reviewer SgUC disputed the core claim that GRPO is effectively outcome-driven SFT, arguing the on-policy distinction is critical. While this is disputed, I do not view it as the essense of the problem (see the discussion above).

- **Lack of Proposed Solutions:** Reviewer UE3A noted the paper identifies problems without offering specific solutions. The authors maintained the paper is a diagnostic critique ("root cause analysis"), leaving it for future work.

**Reviewer Scores:**

Reviewers SgUC and WZko are likely to maintain their score, while reviewers UE3A wmmp are likely to increase the ratings.

---

### Decision · Program_Chairs · 2026-01-26

Reject